# Infectious Inflammatory Processes and the Role of Bioactive Agent Released from Imino-Chitosan Derivatives Experimental and Theoretical Aspects

**DOI:** 10.3390/polym14091848

**Published:** 2022-04-30

**Authors:** Loredana Himiniuc, Razvan Socolov, Vlad Ghizdovat, Maricel Agop, Emil Anton, Bogdan Toma, Lacramioara Ochiuz, Decebal Vasincu, Ovidiu Popa, Viviana Onofrei

**Affiliations:** 1Department of Obstetrics and Gynecology, “Grigore T. Popa” University of Medicine and Pharmacy Iasi, 700115 Iasi, Romania; loredana.himiniuc@umfiasi.ro (L.H.); bogdan.toma@umfiasi.ro (B.T.); 2Department of Obstetrics and Gynecology, Faculty of Medicine, “Grigore T. Popa” University of Medicine and Pharmacy Iasi, 700115 Iasi, Romania; razvan.socolov@umfiasi.ro (R.S.); emil.anton@umfiasi.ro (E.A.); 3Department of Biophysics and Medical Physics, “Grigore T. Popa” University of Medicine and Pharmacy Iasi, 700115 Iasi, Romania; vlad.ghizdovat@umfiasi.ro; 4Department of Physics, “Gheorghe Asachi” Technical University of Iasi, 700050 Iasi, Romania; 5Academy of Romanian Scientists, 050094 Bucharest, Romania; 6Department of Pharmaceutical and Biotechnological Drug Industry, ”Grigore T. Popa” University of Medicine and Pharmacy, 700115 Iasi, Romania; lacramioara.ochiuz@umfiasi.ro; 7Department of Dental and Oro-Maxillo-Facial Surgery, “Grigore T. Popa” University of Medicine and Pharmacy, 700115 Iasi, Romania; decebal.vasincu@umfiasi.ro; 8Department of Emergency Medicine, “Grigore T. Popa” University of Medicine and Pharmacy, 700115 Iasi, Romania; 9Department of Internal Medicine (Cardiology), “Grigore T. Popa” University of Medicine and Pharmacy, 700115 Iasi, Romania; viviana.aursulesei@umfiasi.ro

**Keywords:** chitosan, imine, bioactive agent release, drug delivery systems, fractal/multifractal model, scale transition, intrauterine adhesions, chronic endometritis, endometrial inflammation, viral myocarditis

## Abstract

The paper focuses on the development of a multifractal theoretical model for explaining drug release dynamics (drug release laws and drug release mechanisms of cellular and channel-type) through scale transitions in scale space correlated with experimental data. The mathematical model has been developed for a hydrogel system prepared from chitosan and an antimicrobial aldehyde via covalent imine bonds. The reversible nature of the imine linkage points for a progressive release of the antimicrobial aldehyde is controlled by the reaction equilibrium shifting to the reagents, which in turn is triggered by aldehyde consumption in the inhibition of the microbial growth. The development of the mathematical model considers the release dynamic of the aldehyde in the scale space. Because the release behavior is dictated by the intrinsic properties of the polymer–drug complex system, they were explained in scale space, showing that various drug release dynamics laws can be associated with scale transitions. Moreover, the functionality of a Schrödinger-type differential equation in the same scale space reveals drug release mechanisms of channels and cellular types. These mechanisms are conditioned by the intensity of the polymer–drug interactions. It was demonstrated that the proposed mathematical model confirmed a prolonged release of the aldehyde, respecting the trend established by in vitro release experiments. At the same time, the properties of the hydrogel recommend its application in patients with intrauterine adhesions (IUAs) complicated by chronic endometritis as an alternative to the traditional antibiotics or antifungals.

## 1. Introduction

Chitosan is a biopolymer widely investigated in a large variety of applications due to its outstanding biologic properties [1]. From the chemical point of view, chitosan can be considered a polyamine, the main repeating unit being glucosamine. Thus, the main route for chitosan functionalization is the reaction of the amine functional groups with aldehydes towards imines [2]. A literature survey shows that, in reacting chitosan with various aldehydes, a large variety of materials were prepared, such as coatings [3], films [4], hydrogels [5], and nanoparticles [6]. Their extensive investigation demonstrated an excellent application potential, encouraging researchers to further develop this synthetic route. It was demonstrated that, due to the chitosan particular solubility in acidic medium, the imino-chitosan linkage exists in an equilibrium state between the imine products and reagents [2,5,7]. This apparent drawback was transformed into an advantage by creating dynamic polymer materials capable of responding under the environmental stimuli applied [8,9,10,11]. In this respect, the reversibility of the imino-chitosan derivatives created excellent premises for a controlled release of bioactive aldehydes, where the role of the trigger is played by the water traces in the wet physiological environment.

The deciphering of drug release mechanisms is an important aspect towards the development of new drug delivery systems. Models usually employed in the description of drug release dynamics are based on a differentiable class of models. Examples include: the zero-order model, the Higuchi model, the Hixon- Crowell model, the Korsmeyer-Peppas model, the first order model, etc., i.e., the standard empirical models [12,13,14,15]. Additionally, there also exists a non-differentiable class of models, such as the fractal drug release model [16]. 

Recently, a new class of models for the description of physical systems dynamics and, particularly, of drug release dynamics has gained prominence [17,18,19]. This class of models is based on the Scale Relativity Theory [18,19]. The hypothesis underlining the Scale Relativity Theory with application in the description of drug delivery dynamics is that the structural units of any polymer–drug system move on continuous and non-differentiable curves (fractal/multifractal curves, i.e., three-dimensional fractured/multifractured lines, their non-linearity being dependent on and proportional to the number of interactions within the polymer–drug system). In this context, the fractalization/multifractalization degree will be defined as a measure of the polymer–drug system complexity and physical variables. Therefore, the polymer–drug dynamics will be characterized by fractal/multifractal functions dependent on both spatial and temporal coordinates and also on resolution scales. Consequently, the drug release dynamics become operational not only in the usual space-time but also in the scale space. Because the release properties are dictated only by the intrinsic properties of the polymer–drug complex system, and, moreover, these properties are explained only in scale space, in what follows, it will be shown that various drug release dynamics laws can be associated with scale transitions. Moreover, specific drug release mechanisms will be highlighted.

In this context, the development of a release model of the bioactive compounds bonded to chitosan via reversible imine bonds is very important for the understanding of release behavior and the further development of new systems. In our previous study [20] on an imine-chitosan hydrogel, we proposed a mathematical model useful for monitoring the release of bioactive aldehydes covalently bonded to the chitosan by reversible imine linkage, considered as a polymer–drug system. This model was based on the hypothesis that the imine-chitosan system can be assimilated from a mathematical point of view with a multifractal object. Therefore, its dynamics were analyzed in the framework of the Scale Relativity Theory. As a result, two synchronous dynamics, one in the scale space (associated with the fungicidal activity) and the other in the usual space (associated with the antifungal aldehyde release), became operational through Riccati-type gauges. It followed that their synchronicity (reducible to the isomorphism of two SL (2R)-type groups), implies (by means of its joint invariant functions) bioactive aldehyde compound release dynamics in the form of “kink–antikink pairs” dynamics of a multifractal type [20].

The aim of the paper was to develop a theoretical model that explains the drug release dynamics as scale transitions in order to investigate the release of a bioactive aldehyde covalently bonded to chitosan via imine linkages. To achieve this, we develop a drug release law (different than the one from [20]), and we obtain, through various operational procedures, drug release mechanisms of cellular and chanel-type. In such a context, the bioactive aldehyde contains a boronic acid residue, which endows it with strong antifungal activity. By reacting chitosan with this aldehyde, a dynamic hydrogel was obtained, which progressively released the antifungal aldehyde in a physiologic environment due to the reversible character of the imine bond. Thus, the hydrogels showed strong antifungal activity against the planktonic yeasts and biofilm as well.

## 2. Theoretical Design

### 2.1. Scale Space and Drug Release Laws

For a better understanding of our model, we want to recall several concepts that allow for the description of dynamics in the scale space [20]. Now, let it be considered a multifractal function F(x) in the interval x∈[a,b]. This function can be associated with any multifractal variable that describes drug release dynamics in the scale space. In such a context, to the following sequences of values for x:(1)$$x_{a}=x_{0},\;x_{1}=x_{0}+\varepsilon ,\ldots ,\;x_{k}=x_{0}+k\varepsilon ,\;\ldots ,\;x_{n}=x_{0}+n\varepsilon =x_{b}$$
corresponds F(x,ε) as the broken line that connects the points:(2)$$F\left(x_{0}\right),\;\ldots ,\;F\left(x_{k}\right),\;\ldots ,\;F\left(x_{n}\right)$$

This broken line is an ε-scale approximation of F(x).

Let it be considered a different scale and its $\bar{\varepsilon }$-scale approximation of $F\left(x,\bar{\varepsilon }\right)$. Since F(x) is a multifractal function, it is self-similar almost everywhere, and if ε and $\bar{\varepsilon }$ are sufficiently small, the two approximations should lead to approximately the same results. Comparing these two, an infinitesimal increase/decrease dε of ε corresponds to an infinitesimal increase/decrease $d\bar{\varepsilon }$ for $\bar{\varepsilon }$; thus:(3)$$\frac{d\varepsilon }{\varepsilon }=\frac{d\bar{\varepsilon }}{\bar{\varepsilon }}=d\mu$$

In this framework, the transition for the scale ε+dε and dε must be invariant. It is then possible to consider that the infinitesimal transform of the scale is:(4)$$\varepsilon^{\prime }=\varepsilon +d\varepsilon =\varepsilon +\varepsilon d\mu$$

Performing this transform for F(x,ε), it is obtained:(5)$$F\left(x,\varepsilon^{\prime }\right)=F\left(x,\varepsilon +\varepsilon d\mu \right)$$

Then, through (4):(6)$$F\left(x,\varepsilon^{\prime }\right)=F\left(x,\varepsilon \right)+\frac{\partial F}{\partial \varepsilon }\left(\varepsilon^{\prime }-\varepsilon \right)$$
which then yields:(7)$$F\left(x,\varepsilon^{\prime }\right)=F\left(x,\varepsilon \right)+\frac{\partial F}{\partial \varepsilon }\varepsilon d\mu$$

Let it be observed that for an arbitrary fixed $\varepsilon_{0}$, the following equation is found:(8)$$\frac{\partial \ln\left(\frac{\varepsilon }{\varepsilon_{0}}\right)}{\partial \varepsilon }=\frac{\partial \left(\ln\varepsilon -\ln\varepsilon_{0}\right)}{\partial \varepsilon }=\frac{1}{\varepsilon }$$

Thus, Equation (6) can also be written as:(9)$$F\left(x,\varepsilon^{\prime }\right)=F\left(x,\varepsilon \right)+\frac{\partial F\left(x,\varepsilon \right)}{\partial \ln\left(\frac{\varepsilon }{\varepsilon_{0}}\right)}d\mu$$

Finally, it is obtained:(10)$$F\left(x,\varepsilon^{\prime }\right)=\left(1+\frac{\partial }{\partial \ln\left(\frac{\varepsilon }{\varepsilon_{0}}\right)}d\mu \right)F\left(x,\varepsilon \right)$$

The operator:(11)$$\hat{D}=\frac{\partial }{\partial \ln\left(\frac{\varepsilon }{\varepsilon_{0}}\right)}$$
acts as a dilation/contraction operator, depending on the given process. Thus, the invariance of the equations that describe drug release dynamics is expressed through whether one of these equations is changed if the operator is applied, while specifying that the intrinsic variation of the resolution is $\ln\left(\varepsilon /\varepsilon_{0}\right)$.

Thus, the spontaneous breaking of scale invariance of any multifractal variable Q which shall describe drug release dynamics implies the functionality of the following equation:(12)$$\frac{\partial Q}{\partial \ln\left(\frac{\varepsilon }{\varepsilon_{0}}\right)}=P\left(Q\right)$$
where P(Q) is an arbitrary polynomial associated with the variable Q.

Thus, through a convenient choice of both  Q and $\ln\left(\frac{\varepsilon }{\varepsilon_{0}}\right)$ but also of P(Q), it is possible to mime various drug release laws (such as Zero-order model, First-order model, Weibull model, etc.) as spontaneous symmetry breaking of scale invariance.

In what follows, let such a situation be explained, for the Weibull model. Let it be chosen: (13)$$Q\equiv \ln\left(1-\frac{m_{\tau }}{m_{\infty }}\right),\;\;\frac{\varepsilon }{\varepsilon_{0}}=\exp^{\tau^{\beta }},\;\;P\left(Q\right)\equiv -\alpha$$

Then, through integration and the adequate interpretation of the variables and parameters [12], (12) becomes a Weibull law of multifractal type:(14)$$\frac{m_{\tau }}{m_{\infty }}=1-exp^{-\alpha \tau^{\beta }}$$

It is noted that the release parameters α and β are dependent on the multifractality degree of the drug release curves. 

### 2.2. Scale Space and Drug Release Mechanisms

The release dynamics of the polymer–drug complex system can be described by the states function $\Psi_{S}$. According to [18], this function satisfies the Schrödinger-type differential equation from the scale space:(15)$$\lambda_{S}^{2}\frac{\partial^{2}\Psi_{S}}{\partial X_{S}^{2}}+i\lambda_{S}\frac{\partial \Psi_{S}}{\partial \tau }=0\;$$
where $X_{S}=\ln\Lambda$ is the spatial scale coordinate, τ is the temporal scale coordinate, and $\lambda_{S}$ is the diffusion coefficient associated with the fractal–non-fractal transition. Now, by also employing the differential Schrödinger-type equation from the scale space for the complex conjugate ${\bar{\Psi }}_{S}$, based on the procedure from [18,19], the states density conservation law in the scale space can be obtained:(16)$$\frac{\partial \rho_{S}}{\partial \tau }+\frac{\partial J_{S}}{\partial X_{S}}=0$$
where
(17)$$\rho_{S}=\Psi_{S}{\bar{\Psi }}_{S},\;\;J_{S}=i\lambda_{S}\left(\Psi_{S}\frac{\partial {\bar{\Psi }}_{S}}{\partial X_{S}}-{\bar{\Psi }}_{S}\frac{\partial \Psi_{S}}{\partial X_{S}}\right)$$

In (17), $\rho_{S}$ is the states density in the scale space and $J_{S}$ is the current density in the scale space.

Considering the meaning of the state function $\Psi_{S}$ by means of a probability density $\rho_{S}=\Psi_{S}{\bar{\Psi }}_{S}$, in what follows, non-manifest dynamics states in complex systems, through metrics of the Lobachewski plane, will be generated. Indeed, let it be admitted the functionality of the relation:(18)$$x^{2}+y^{2}=1$$
where
(19)$$\Psi_{S}=a+ib,\;\;x=\frac{a}{\rho_{S}},\;\;y=\frac{b}{\rho_{S}}$$

In these conditions, the metric of the Lobachewski plane can be produced as a Caylean metric of a Euclidean plane, for which the absoluteness is a circle with unit radius (18). Thus, the Lobachewski plane can be put into a biunivocal correspondence with the interior side of the circle. The general procedure of metrization of a Caylean space starts with the definition of the metric as an anharmonic ratio (for details, see [19]). As such, let it be supposed that the absoluteness of the space is represented by the quadratic form Ω(X,Y), where X denotes any vector. The Cayleyan metric is then given by the differential quadratic form:(20)$$\frac{-ds^{2}}{k^{2}}=\frac{\Omega \left(dX,dX\right)}{\Omega \left(X,X\right)}-\frac{\Omega^{2}\left(X,dX\right)}{\Omega^{2}\left(X,X\right)}$$
where Ω(X,Y) is the duplication of Ω(X,Y) and *k* is a constant connected to the space curvature.

In the case of the Lobachewski plane, there is:(21)$$\Omega \left(X,X\right)=1-x^{2}-y^{2}\;\Omega \left(X,dX\right)=-xdx-ydy\;\Omega \left(dX,dX\right)=-dx^{2}-dy^{2}$$
which yields
(22)$$\frac{-ds^{2}}{k^{2}}=\frac{\Omega \left(dX,dX\right)}{\Omega \left(X,X\right)}-\frac{\Omega^{2}\left(X,dX\right)}{\Omega^{2}\left(X,X\right)}$$

Performing the coordinate transformation:(23)$$x=\frac{h\bar{h}-1}{h\bar{h}+1},\;y=\frac{h+\bar{h}}{h\bar{h}+1}$$
the metric (22) becomes the Lobachewski metric:(24)$$\frac{-ds^{2}}{k^{2}}=-4\frac{dhd\bar{h}}{{\left(h-\bar{h}\right)}^{2}}$$

It is possible to observe that the absoluteness $1-x^{2}-y^{2}=0$ tends to the straight line Im h=0. The straight lines of the Euclidean plane tend to circles with centers located on the real axis of the complex plane (h).

Now, let it be considered that the complex system dynamics are described by the variables $\left(Y^{j}\right)$, for which the following multifractal metric was discovered:(25)$$h_{ij}dY^{i}dY^{j}$$
in an ambient space of the multifractal metric:(26)$$\gamma_{\alpha \beta }dX^{\alpha }dX^{\beta }$$

In this situation, the field equations of the complex system dynamics are derived from a variational principle connected to the multifractal Lagrangian:(27)$$L=\gamma^{\alpha \beta }h_{ij}\frac{dY^{i}dY^{j}}{\partial X^{\alpha }\partial X^{\beta }}$$

In the current case, (25) is given by (24), the field multifractal variables being h and h¯ or, equivalently, the real and imaginary part of *h*. Therefore, if the variational principle:(28)δ∫ Lγd3x
is accepted as a starting point where γ=|γαβ|, the main purpose of the complex system dynamics research would be to produce multifractal metrics of the multifractal Lobachewski plane (or relate to them). In such a context, the multifractal Euler equations corresponding to the variational principle (31) are:(29)(h−h¯)∇(∇h)=2(∇h)2(h−h¯)∇(∇h¯)=2(∇h¯)2
which admits the solution:(30)h=(cosh(χ2)−sinh(χ2))e−iα(cosh(χ2)+sinh(χ2))e−iα, α∈ℝ
with α real and arbitrary, as long as (χ2) is the solution of a Laplace-type equation for the free space, such that ∇2(χ2)=0. For a choice of the form α=2ωt, in which case a temporal dependency was introduced in the complex system dynamics, (30) becomes:(31)h=i[e2χsin(2ωt)−sin(2ωt)−2ieχ]e2χ[cos(2ωt)+1]−cos(2ωt)+1

We present in Figure 1a,b two distinct self-structuring modes of the polymer–drug system in the release process: channel-type self-structuring modes in Figure 1a, and cellular-type self-structuring modes in Figure 1b. In our opinion, the channel-type self-structuring modes can describe the release process induced by weak interactions between the drug and matrix, while the cellular-type self-structuring modes can describe the release process induced by strong interaction between the drug and matrix.

## 3. Experimental Design

The mathematical model has been validated on two hydrogels based on chitosan and 2-formylphenylboronic acid, prepared by imination reaction of the two components in different molar ratios, to give two hydrogels containing 0.071% (coded H0.071) and 0.142% (coded H0.142) boronic aldehyde [7]. The properties of the hydrogels affecting the drug release were briefly presented in the Results and Discussions section.

The development of the mathematical model considered the release dynamic of the aldehyde in the scale space.

## 4. Results and Discussion

A mathematical model, considering drug release laws in scale space and various drug release mechanisms (of cellular and channel-type), was developed in order to better understand the release of the bioactive compounds covalently bonded to chitosan via reversible imine linkage (Appendix A) [7,20]. The model has been validated by fitting on the in vitro release curves of two hydrogels prepared by acid condensation reaction via the formation of reversible imine bonds and the supramolecular ordering of newly formed imine units (Figure 2a) [7]. The supramolecular architecting of the hydrogels was confirmed by polarized optical microscopy, which displayed strong birefringence reminiscent of a banded texture, indicating that the newly formed imine units bonded to the chitosan formed ordered clusters playing the role of non-classical crosslinking nodes (Figure 2b) [5,7,21]. The hydrogels had a porous microstructure with pores diameter around 30 µm and relative thick pore walls of 2–5 µm, suitable for a good transport of bioactive aldehyde during the release process (Figure 2c) [22,23].

The hydrogels showed a pH-dependent swelling behavior with a fast swelling in acidic media of pH = 4.2 characteristic to vaginal media (mass equilibrium swelling (MES) ≈ 27), a moderate one in neutral medium of pH = 7 (MES ≈ 11), and a low swelling in medium of pH 7.4 characteristic to the intraperitoneal environment (MES ≈ 2.5). This indicates an excellent ability of the hydrogels to assure a moisture medium in different parts of the body characterized by different pH and moreover indicate their propensity to be used for the treatment of endometriosis by intraperitoneal administration [24]. The hydrogels showed excellent antifungal activity against the Candida species by killing both planktonic yeasts and biofilms in 24 h [7]. The in vitro release curves showed that the hydrogel H0.142 released 75% of aldehyde in 24 h, while H0.071 released 98% (Figure 3). Looking at the release profile of the two samples, it can be seen that there is a burst release in the first four hours followed by a progressive release of aldehyde in time, most probably correlated with the swelling rate. It can be envisioned that the progressive swelling supported the shifting of the imination equilibrium of the reagents and the diffusion of the released aldehyde. Thus, the faster sink of the aldehyde from the hydrogel to the release medium assured good conditions for the release of new aldehyde amounts. This explains the slower release from the hydrogel containing higher amount of aldehyde, for which the shifting of the imination to the reagents is hindered by the higher concentration of free aldehyde.

From the fitting of the in vitro release data using the classical drug release model, it is difficult to deduce with a high degree of trust the mechanism of the drug release, as the release mechanism is influenced by a high number of factors, especially taking into account the difference in the spatial and temporal scales at which various aspects of the drug release process are investigated. The theoretical model developed here can validate through an adequate calibration of the experimental data and by choosing the constants according to the particularities of our polymer–drug system. The main characteristic of our model is the fractalization degree, which, according to some of our previous results, is the main driving factor behind the individual dynamic of the structural units (for drug release, see [25], or, for other physical processes such as fibre systems dynamics, see [26]). More precisely, referring to physical processes, in the previously mentioned papers, various aspects of the dynamics of complex systems (such as physical processes in complex fluids, in condensed matter, in quantum mechanics, in nanostructures, in fields theory, etc.) are highlighted. Taking into account the complexity of such processes, explaining them through the choice of a convenient fractality degree (i.e., both the scale resolution and the fractal dimension of the motion curve) entails a “scanning” of the implied dynamics. The calibration process that can lead to the estimation of the fractalization degree is not a trivial one, as it strongly depends on the nature of the investigated phenomena. This calibration technique was previously successfully tested for other physical phenomena, such as plasma oscillations or the transient dynamics of laser producing plasma translated into fractal movement in the framework of NSRT, with promising results [27]. We can observe that the model fits well in both cases; the H0.017 case reaches a high increase slope for small scale moments of time, while the H0.142 has a lower release rate. The addition of hydrogels containing boronic aldehyde leads to an increase in the overall fractality degree of the polymer–drug system by a factor of 3.2 and a decrease in the overall released drug mass by a factor of 1.3 in the saturation region. This understanding of the fractality degree as a measure of the expansion dynamics and velocity is in line with our view from [1] and expands the vision already cemented in [17]. The medium becoming of a high degree of fractality results in the increase in the number of structural units (particles, molecules, clusters, monomers, etc.) and therefore decreases the energy and kinetic of the system. Different drug–polymer configurations will further be tested to expand the horizon of the multifractal model and to better understand the role of fractality in the biomolecular interaction and various interaction scales.

## 5. Perspectives in Infectious Inflammatory Processes

In the following, we want to highlight some possible implications of the bioactive agent released from imino-chitosan derivatives in some infectious inflammatory processes.

Let us first note that Asherman`s syndrome (AS) is defined by endometrial damage and the existence of intrauterine adhesions (IUAs). Many causes may determine fibrotic tissue replacement within the endometrium and adhesions formation between the uterine walls, leading to uterine cavity deformation. The most incriminated causes represent mechanical trauma, non-traumatic factors (infection, puerperal and postabortal sepsis), fibrosis imbalance, the increase of cytokines expression, and neovascularization. These explain the role of inflammation-mediated endometrial injury in both AS and IUAs pathogenesis. It was found that chronic endometritis (CE) impacts endometrial repair and contributes to the IUAs recurrence by preserving local hypoxia and increasing inflammation. Therefore, CE influences matrix metalloproteinases-9 (MMP-9) and tumor growth factor-β1 (TGF-β1) expression, impairing the homeostasis of the extrauterine matrix and leading to exacerbated fibrosis [28]. The nuclear factor-kappa B (NF-kB) transcription factor plays an important role in the pathogenesis of IUA by being considered a critical regulatory pathway that manages the production of cytokines and cell survival. The activation of the NF kB signaling pathway may increase the expression of TGF-β and later the expression of connective tissue growth factor (CTGF/CCN2), finally leading to adhesiogenesis by fibrotic processes promotion [29,30]. The role of the ADAM family of proteins in intrauterine adhesiogenesis is already known. Normally, the ADAM family disintegrates the extracellular matrix (ECM) anchoring proteins that are physiologically regulated, but, in AS, a high expression of ADAM-15 and ADAM-17 was found within the endometrium, which reveals its implication in adhesions formation. Moreover, other factors may contribute to IUAs genesis, such as the imbalance between TGF β1 and small mother against decapentaplegic 3 (SMAD3) signaling (abnormally upregulated) and SMAD7 signaling (downregulated) [31].

Chronic endometritis is usually presented as an asymptomatic inflammatory pathology of the endometrium. During the reproductive cycle, the structure of the endometrium is significantly changed, preserving a relatively unmodified environment. Still, the endometrium presents susceptibility to various fungal and bacterial agents that may determine chronic endometritis and later create the perfect inflammation-mediated endometrial injury environment for IUA formation. The endometritis features are inflammation, tissue impairment, and necrosis, which are developed and maintained under the expression of inflammatory mediators such as prostaglandin E2 (PGE2), nitric oxide (NO), interleukin-1β (IL-1β), interleukin-6 (IL-6), interleukin-8 (IL-8), and damage-associated molecular patterns (DAMPs). On the other side, MMPs have been correlated with endometrial remodeling, but the mechanisms beyond are still poorly understood [32]. It is suggested that patients with IUAs complicated with CE present a negative impact of steady-state imbalance of endometrial fibrosis and exacerbate endometrial fibrosis, an endometrial environment that promotes the formation and recurrence of IUA, affecting endometrial receptivity with great negative impact on the patient’s fertility. 

Animal studies have proved chitosan antibacterial and antifungal properties, as well as anti-adhesion and hemostasis features, suggesting its potential to decrease inflammation and fibrosis in endometrial tissue, making it a possible alternative for traditional antibiotics [33,34]. Moreover, its advantage has been reported in the vaccine against Coxsackievirus B3 (CVB3)-induced myocarditis. Coxsackievirus B3 infection represents one of the most frequent causes of viral myocarditis that has no available vaccine. Chitosan-p high mobility group box1 (HMGB1 or amphoterin) coimmunization raises the level of CVB3-specific fecal secretory immunoglobulin A (SigA), enhances mucosal T cell proliferation and IFN-γ-secreting cell production, and provides higher anti-viral protection compared to the control group. The capacity of pHMGB1 to recruit and stimulate the maturation of dendritic cells may increase the immunity of the chitosan-pHMGB1 complex, leading to enhanced mucosal immune responses as well as alleviating viral myocarditis [35]. Chai et al. (2014) have shown that using the cytosolic DNA sensor AIM2 as a mucosal auxiliary may increase the immunogenicity of chitosan-pVP1 DNA vaccine and mucosal immune response to offer great protection against the virus [36].

## 6. Conclusions

A multifractal theoretical model of drug release dynamics in scale space was developed and validated by fitting on the in vitro release curves of an antimicrobial aldehyde covalently bonded to chitosan via imine linkages to form hydrogels. This model is an extension of our previous results, published in [20]. In addition to our findings from [20], we have obtained conservation laws in scale space and also specified the types of drug release mechanisms.

This system has been particularly interesting due to its properties, such as controlled release of the antimicrobial agent triggered by the imine reversibility and strong antimicrobial activity against planktonic yeasts and biofilms, which indicate these systems as suitable for application in the treatment of chronic inflammations, those affecting the endometrial receptivity that may impact the reproductive prognosis being especially envisaged. 

The fitting of the new developed model demonstrated that the hydrogels containing the antimicrobial aldehyde leads to an increase in the overall fractality degree of the polymer–drug system by a factor of 3.2 and a decrease in the overall released drug mass by a factor of 1.3 in the saturation region. The successful application of the model to this particular system encourages its further application to other systems in which a bioactive compound is bonded to the chitosan via reversible imine linkage.

## Figures and Tables

**Figure 1 polymers-14-01848-f001:**
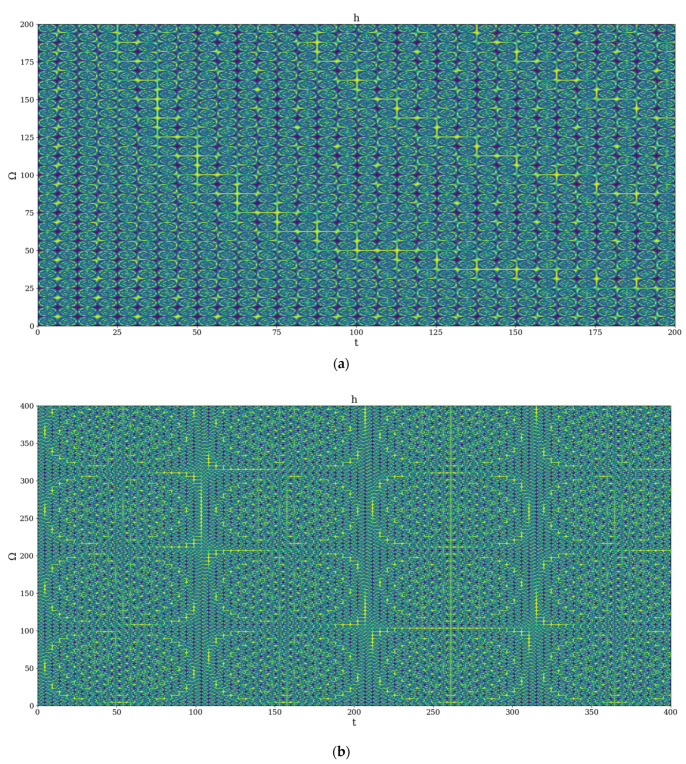
2D dynamics at global scale resolution plot of |h| from (31) for χ=2.35 and (**a**) Ω=0−200, t=0−200, (**b**) Ω=0−400, t=0−400.

**Figure 2 polymers-14-01848-f002:**
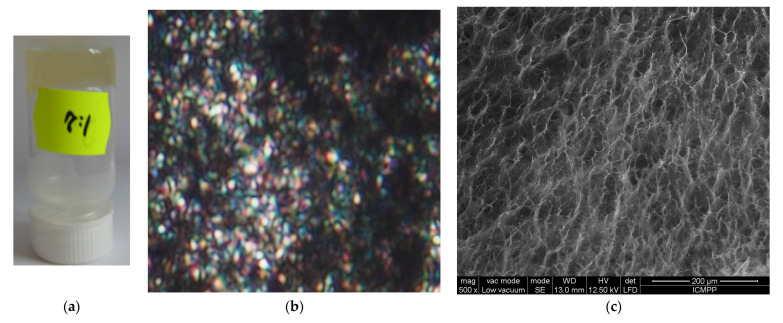
Representative images of the (**a**) hydrogels, (**b**) POM texture, and (**c**) SEM microstructure.

**Figure 3 polymers-14-01848-f003:**
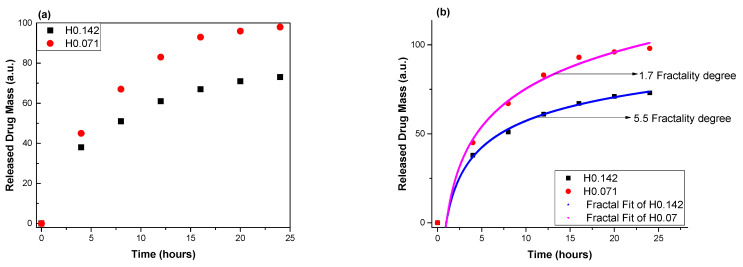
The in vitro release curves of the bioactive aldehyde from chitosan-based hydrogels (**a**) and the multifractal fit of the empirical data (**b**).

## Data Availability

The research data used to support the findings of this study are included within the article.

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
