# Peer review of "Infectious Inflammatory Processes and the Role of Bioactive Agent Released from Imino-Chitosan Derivatives Experimental and Theoretical Aspects"

_polymers, 2022, doi:10.3390/polym14091848_

Round 1

Reviewer 1 Report

Title: Infectious inflammatory processes and the role of bioactive agent released from imino-chitosan derivatives. Experimental and theoretical aspects

Authors: Loredana Himiniuc , Razvan Socolov , Vlad Ghizdovat , Maricel Agop, Emil Anton , Bogdan Toma , Lacramioara Ochiuz , Decebal Vasincu , Ovidiu Popa , Viviana Onofrei  

-This paper contains two part one is experimental part in which two bioactive imino-chitosan derivatives are prepared by acid condensation reaction of the chitosan amine groups with an aldehyde containing a boronic acid residue with strong antifungal activity, although, the synthesis and strong antifungal activities of the imino-chitosan derivatives containing that particular aldehyde have already been reported earlier with having one common author.

-The second part is the development of a theoretical model which is rather new.   

My main concern is with the experimental part and the novelty of this part should be discussed in detail by discussing the previously reported works on the topic (Ref# 22: Imino-chitosan biopolymeric films. Obtaining, self-assembling, surface and antimicrobial properties and Ref # 27: Dual crosslinked iminoboronate-chitosan hydrogels with strong antifungal activity against Candida planktonic yeasts and biofilms). How the present works, particularly, the experimental part advances the previously established research. If it is related to the comparison of only the release pattern of two different derivatives (the synthesis, characterization and therapeutic properties of that material is already reported) it is not suitable for publication because it does not provide sufficient information to be published as a full article and require more work like entrapment of any other relevant drug and its simultaneous release with the antifungal aldehyde along with the efficacy of the released drug.   

Minor points:

-There is no need to write the first five lines of the abstract section and these can be deleted because this information is given in the introduction section. 

-The resolution of the images given in Figure 1 should be improved. 

Reviewer 2 Report

The materials used in this study and part of the results provided are already published by the same author in “Marin, L.; Popa, M.; Anisiei, A.; Irimiciuc, S.-A.; Agop, M.; Petrescu, T.-C.; Vasincu, D.; Himiniuc, L. A Theoretical Model for Release Dynamics of an Antifungal Agent Covalently Bonded to the Chitosan. Molecules 2021, 26, 2089. https:// doi.org/10.3390/molecules26072089” Hence, I would not recommend this paper once again publish in “Polymers”.

  1. The method of preparation and the materials are exactly same.
  2. The same results were repeated with slight modification on sample names only.
  3. The targeted applications also same.
  4. The original paper was not cited anywhere in the manuscript

Reviewer 3 Report

The presented article contains a very large amount of literature information and a limited amount of experimental material. In my opinion, for a publication in the Q1 journal, the article must be supplemented with new experimental results.

In the abstract, the authors state: "assuring efficient antifungal activity along 48 hours." (Line 35-36) as also “The imino-chitosan derivates showed excellent antifungal activity against Candida species.” (Line 38). It is not clear Is this the authors' own data or data from a previously published article? (Carbohydrate Polymers//2016. V. 152. P. 306.) If these are the authors' own data, then there is no corresponding section on the study of antifungal activity in the body of the article. If this is previously published data, then these sentenses should be excluded from the abstract.

The introduction contains a significant fragment devoted to the pathogenesis of Chronic endometritis and methods of their therapy. However, further, the authors do not provide any experiments confirming the effectiveness of the tested chitosan derivative in the treatment of endometritis.

Reference 26 line 153 is incorrect.

Line 224 – 225: "hydrogels containing an antifungal aldehyde were prepared by acid condensation via formation of reversible imine bonds. " The formation of the resulting compound must be confirmed, for example IR spectroscopy or NMR.

Round 2

Reviewer 1 Report

Authors have somewhat revised the manuscript, however, as one of my fellow reviewer has also noticed that the materials used in this study and part of the results provided are already published by the same authors in their another publication, (https:// doi.org/10.3390/molecules26072089) the novelty of the work still a concern. The authors must give the clear explanation by citing this reference in introduction section rather giving the one or two sentence only in conclusion section. 

In response to the reviewer, the authors must specify exactly what change is and where the change is made in the revised manuscript (strictly define the lines in the revised manuscript).

Reviewer 2 Report

Before acceptance, the authors should completely investigate the prepared hydrogels. IR, swelling, and mechanical properties must be studied.

The experimental part should be outlined in depth.

I think, There are many articles related to the target of this manuscript. Thus, the authors need to outline their targets and the difference between their work and the published articles.

Reviewer 3 Report

Unfortunately, in the revised version and in response to the comments of the reviewers, the authors did not provide convincing new data. The manuscript does not contain enough new information to be published in Polymers.

Round 3

Reviewer 3 Report

The authors have significantly revised the manuscript. The comments of the reviewer were taken into account. Regarding the preparation, characterization and in vitro release of aldehyde sections from the hydrogel, I have no comments.